# Low-Temperature Solution-Processed HfZrO Gate Insulator for High-Performance of Flexible LaZnO Thin-Film Transistor

**DOI:** 10.3390/nano13172410

**Published:** 2023-08-25

**Authors:** Yeoungjin Chang, Ravindra Naik Bukke, Jinbaek Bae, Jin Jang

**Affiliations:** 1Advanced Display Research Center, Department of Information Display, Kyung Hee University, Seoul 02447, Republic of Korea; yjchang@gachon.ac.kr (Y.C.); jbbae@tft.khu.ac.kr (J.B.); 2Department of Semiconductor Display, Gachon University, Seongnam-si 13120, Republic of Korea; 3School of Mechanical & Materials Engineering, Indian Institute of Technology Mandi, Mandi Pradesh 175075, India

**Keywords:** flexible, hafnium zirconium oxide, lanthanum zinc oxide, solution-processed, spray pyrolysis, thin-film transistor

## Abstract

Metal-oxide-semiconductor (MOS)-based thin-film transistors (TFTs) are gaining significant attention in the field of flexible electronics due to their desirable electrical properties, such as high field-effect mobility (μ_FE_), lower I_OFF_, and excellent stability under bias stress. TFTs have widespread applications, such as printed electronics, flexible displays, smart cards, image sensors, virtual reality (VR) and augmented reality (AR), and the Internet of Things (IoT) devices. In this study, we approach using a low-temperature solution-processed hafnium zirconium oxide (HfZrOx) gate insulator (GI) to improve the performance of lanthanum zinc oxide (LaZnO) TFTs. For the optimization of HfZrO GI, HfZrO films were annealed at 200, 250, and 300 °C. The optimized HfZrO-250 °C GI-based LaZnO TFT shows the μ_FE_ of 19.06 cm^2^V^−1^s^−1^, threshold voltage (V_TH_) of 1.98 V, hysteresis voltage (V_H_) of 0 V, subthreshold swing (SS) of 256 mV/dec, and I_ON_/I_OFF_ of ~10^8^. The flexible LaZnO TFT with HfZrO-250 °C GI exhibits negligible ΔV_TH_ of 0.25 V under positive-bias-temperature stress (PBTS). The flexible hysteresis-free LaZnO TFTs with HfZrO-250 °C can be widely used for flexible electronics. These enhancements were attributed to the smooth surface morphology and reduced defect density achieved with the HfZrO gate insulator. Therefore, the HfZrO/LaZnO approach holds great promise for next-generation MOS TFTs for flexible electronics.

## 1. Introduction

Metal oxide semiconductors (MOS) are gaining significant interest as channel layers in thin-film transistors (TFTs), particularly for active-matrix light-emitting diode (AMOLED) displays [1,2,3]. These materials exhibit desirable electrical properties, such as high mobility, near-zero threshold voltage, low off-state current, excellent area uniformity, reliability, and cost-effective mass production. MOS-based TFTs hold the potential to supplant conventional polycrystalline-Si (Poly-Si) or amorphous-Si (a-Si) TFTs across diverse applications. Their appeal lies in offering higher performance characteristics, including improved carrier mobility and lower off-state leakage. Moreover, MOS-based TFTs can be fabricated via cost-effective and scalable processes, making them a viable candidate for next-generation electronic devices. As a result, these TFTs present a compelling alternative for enhancing electronic technologies and fostering innovation in various industries [1,2,3]. The fabrication of MOS TFTs can be achieved through either solution [1,2,3,4,5] or vacuum process [1,6]. Solution processes [1,2,3,4,5], including spray pyrolysis, spin coating, and inkjet printing, are commonly employed for MOS TFT fabrication [7,8,9,10,11,12,13,14,15,16,17,18,19]. Solution processes offer the advantage of operating at low temperatures, facilitating the deposition of metal-oxide-semiconductor (MOS) films on flexible substrates. This characteristic is essential for the development of flexible electronics as it avoids substrate damage and allows for the fabrication of bendable and stretchable devices [1]. In solution-processed oxide TFTs, high-k dielectric materials, for example, aluminum oxide (AlOx) [12,20], hafnium oxide (HfOx) [21], zirconium oxide (ZrOx) [2,10,17,22], aluminum zirconium oxide (AlZrOx) [13,23], boron-doped ZrOx (BZrO) [9], hafnium zirconium oxide (HfZrOx) [24], and lanthanum-doped ZrOx (LaZrO), are employed to achieve high-performance devices. While other MO semiconductors, such as indium gallium tin oxide (IGTO) [25,26] and indium gallium zinc oxide (IGZO) [5] TFTs, typically utilize a SiO_2_ gate insulator, solution-processed oxide TFTs offer the advantage of utilizing high-k dielectrics for improved performance [8].

The utilization of high-k gate dielectric materials through solution processes holds tremendous promise in realizing high-performance thin-film transistors (TFTs). These materials offer advantageous electrical properties, such as higher dielectric constants, enabling enhanced control of charge carriers in the transistors. Additionally, the solution-based approach allows for cost-effective and scalable fabrication, making it a viable option for advancing next-generation electronic devices with improved efficiency and functionality. J. Li et al. report ZTO/AlZrOx TFT with μ_sat_ of 12.5 cm^2^V^−1^s^−1^, V_TH_ of 0.3 V, I_ON_/I_OFF_ of 8 × 10^7^, and SS of 0.150 V/dec [27]. Tue et al. demonstrated the use of LaZrO gate dielectric in ZrInZnO TFTs, resulting in a saturated mobility (μ_sat_) of 6.23 cm^2^V^−1^s^−1^, I_ON_/I_OFF_ of 10^9^, and SS of 0.19 V/dec [8]. Park et al. report In_2_O_3_/ZrO_2_:B TFT with μ_sat_ of 39.3 cm^2^V^−1^s^−1^, V_TH_ of 2.46 V, I_ON_/I_OFF_ of 10^7^, and SS of 0.263 V/dec [9]. When considering practical applications, TFTs are fabricated on flexible plastic substrates, such as polyethylene naphthalate (PEN), polyethylene terephthalate (PET), and polyimide (PI). These plastic substrates offer mechanical flexibility, making them ideal candidates for foldable or flexible electronics and displays. Consequently, a low-temperature process becomes essential for compatibility with these substrates. Numerous research groups have focused on developing doped ZrOx gate insulators to enhance the performance of flexible metal oxide TFTs [2,22].

In this study, we present the utilization of a low-temperature solution-processed hafnium zirconium oxide (HfZrOx) GI to achieve high-performance LaZnO TFTs. The HfZrO films were carefully annealed at a temperature of 200, 250, and 300 °C, respectively, for optimization. The optimized HfZrO-250 °C GI-based LaZnO TFT demonstrates an impressive field-effect mobility (μ_FE_) of 19.06 cm^2^V^−1^s^−1^, a low threshold voltage (V_TH_) of 1.98 V, and an exceptionally sharp subthreshold swing (SS) of 256 mV/dec. Notably, the device exhibits zero hysteresis voltage (V_H_) and an outstanding I_ON_/I_OFF_ ratio of ~10^8^, ensuring high-performance transistor operation. Furthermore, the flexible HfZrO-250 °C/LaZnO TFT exhibits remarkable stability under positive-bias-temperature stress (PBTS) with minimal ΔV_TH_, indicating the reliability of the device over the extended operation. These exceptional enhancements can be attributed to the smooth surface morphology achieved by the HfZrO GI and the reduced defects at the interface between the HfZrO gate insulator and the oxide semiconductor (LaZnO). Our findings establish the HfZrO/LaZnO approach as a highly promising avenue for developing flexible oxide TFTs, especially for next-generation flexible displays. The novel combination of materials and the low-temperature solution processing offer significant potential for advancing flexible electronics.

## 2. Materials and Methods

To prepare the HfZrO precursor solution, we dissolved zirconyl chloride hydrated (ZrOCl·8H_2_O) and hafnium chloride (HfCl_4_) in a mixture of ethylene glycol (65%) and acetonitrile (35%). The resulting solution, 20 mL in volume, was transferred to a 0.250 L flask. The temperature was steadily increased from room temperature (RT) to 95 °C in 10 °C increments. The solution was maintained at 95 °C for 10 min, resulting in the formation of a transparent precursor solution after cooling to RT. For the synthesis of the 0.2 M LaZnO precursor solution, zinc acetate dihydrate, lanthanum (III) nitrate hexahydrate, and ammonium acetate were added to a solvent called 2-Methoxyenthonal (2ME). The stoichiometry of HZO and LaZnO films are H_0.10_Z_0.90_O and La_0.10_Zn_0.90_O, respectively. The detailed method for preparing the HZO and LaZnO precursor solution can be found elsewhere [2,24]. The precursor solutions were stirred for 2 h under an N2 environment to ensure homogeneity. Finally, a 0.45 µm polytetrafluoroethylene (PTFE) filter was employed to obtain particle-free precursor solutions.

The HfZrO film was deposited via spin coating onto a glass substrate at room temperature in the ambient environment. After deposition, the sample was placed on a hot plate at 140 °C for 5 min and subsequently annealed in an air furnace at temperatures ranging from 200 to 300 °C. The resulting HfZrO thin films were labeled as HfZrO-200 °C, HfZrO-250 °C, and HfZrO-300 °C, corresponding to the annealing temperatures. The LaZnO film, on the other hand, was deposited using spray pyrolysis onto a glass substrate at a substrate temperature of 350 °C.

In our fabrication process, we employed a bottom gate and top contact configuration for the LaZnO TFTs. Initially, a 40 nm molybdenum (Mo) film was sputtered onto the substrate and patterned to create the gate electrodes. Following the deposition of the HfZrO film, the sample was subjected to a 5 min treatment on a hotplate at 140 °C. The LaZnO solution was then deposited onto the substrate at a temperature of 350 °C using spray pyrolysis. The thickness of the LaZnO films was measured using the Alpha-Step D-500 Stylus Profiler (D-500 Stylus Profiler, KLA Instrument, Hayward, CA, USA). The LaZnO layer was patterned using conventional photolithography to form the active island. Finally, a 40 nm thick Mo layer was sputtered and patterned to create the source/drain (S/D) electrodes.

For the fabrication of flexible LaZnO TFTs, we employed a bottom gate and top contact (BGTC) structure on a polyimide (PI) substrate. Initially, a thin layer of carbon nanotube‒graphene oxide (CNT:GO) composite was deposited through spray pyrolysis at 100 °C [3,8]. Subsequently, a 10 μm thick PI layer was spin-coated onto the CNT:GO layer and annealed for 2 h under a nitrogen (N_2_) atmosphere. To provide a gas barrier, a SiNx/SiOx buffer layer with a total thickness of 125 nm (25 nm for each layer) was deposited. The LaZnO channel layer was then deposited at a substrate temperature of 350 °C via spray pyrolysis. The LaZnO active islands were patterned. Finally, a 40 nm molybdenum (Mo) layer was sputtered and patterned to create the source/drain (S/D) contacts. For a detailed fabrication process flow of the LaZnO TFT, refer to the appropriate literature [2,17].

To characterize the HfZrOx films, we conducted UV-visible spectroscopy (transmittance and absorbance) using a Scinco S-4100 instrument. The film thickness was measured with an Alpha step, while the refractive index was determined using ellipsometry. Surface morphology (including RMS roughness) was examined using atomic force microscopy (AFM). Chemical composition and elemental analysis of the metal oxide films were studied through X-ray photoelectron spectroscopy (XPS) with a PHI 5000 Versa Probe (PHI 5000 Versa Probe, Ulvac-PHI, Chigasaki, Japan) under pressure of 7.5 × 10^−5^ mTorr.

The electrical properties of TFTs were measured using an Agilent 4156C semiconductor parameter analyzer. The measurements were conducted at room temperature under dark conditions. The V_TH_ was determined by employing the linear extrapolation method on the (I_DS_)^1/2^ vs. V_GS_ plot, using the *x*-axis intercept. The μ_sat_ was obtained from the linear section of the (I_DS_)^1/2^ vs. V_GS_ curve. The SS was determined from the linear region of the log (I_DS_) vs. V_GS_ fit using Formula (2).
(1)$$I_{DS}=\frac{1}{2}\frac{W}{L}{\mu_{sat}C}_{ox}{{(V}_{GS}-V_{TH})}^{2},$$
(2)$$SS=\frac{{dV}_{GS}}{{d(log\;I\;}_{DS)}}$$
where I_DS_, W/L, μ_sat_, C_ox_, V_TH_, and V_GS_ are the drain current, channel width, channel length, saturation mobility, gate oxide capacitance, threshold voltage, and gate voltage, respectively.

## 3. Results and Discussion

Figure 1a illustrates the process flow for fabricating the HfZrOx thin film on a glass substrate. Further details can be found in the Materials and Methods Section. In Figure 1b, it is evident that the HfZrOx thin film exhibits transmittances exceeding 90% in the visible region. The optical band gaps of HfZrO-200 °C, HfZrO-250 °C, and HfZrO-300 °C thin films are shown in Figure 1c. For the UV-Vis experiment, the HfZrO_x_ films were deposited on the glass, where the absorption of a glass substrate can be negligible [2]. The band gap (E_g_) of HfZrO-250 °C is 5.85 eV, as can be seen in Figure 1c [2,28,29,30,31]. The E_g_ was extracted using the formula: αhν = A (hν − E_g_)^1/n^, where hν, n, A, and α are the photon energy, optical transition exponent, a proportionality constant in the absorption process, and absorption coefficient, respectively. The refractive index (R.I.) of HfZrO_x_ films was measured by ellipsometry. Figure 1d displays the refractive indices (RI) of HfZrO-200 °C, HfZrO-250 °C, and HfZrO-300 °C, with corresponding values of 1.68, 1.79, and 1.83, respectively. The increase in film density is evident from the RI values observed in the visible range. In metal oxide thin film, the composition of the film density can be correlated with RI values. As the annealing temperature increases, the film density of HfZrOx also increases. A lower RI value indicates an increase in film porosity, which subsequently affects both the optical and electrical properties [2,17,28,29,30].

Figure 2 illustrates the frequency-dependent capacitance of the HfZrO-200 °C, HfZrO-250 °C, and HfZrO-300 °C GIs. The capacitance was measured with a frequency from 20 Hz to 2 MHz. The capacitance values of HfZrO-200 °C, HfZrO-250 °C, and HfZrO-300 °C are 384, 455, and 476 nF/cm^2^, respectively. In the capacitance curve of the HfZrOx film annealed at 200 °C, a decrease in capacitance is observed from around 10 kHz, while the capacitance remains constant up to 300 kHz for HfZrOx-300 °C, with degradation commencing at 300 kHz. This behavior is attributed to the lower defect density present in the HfZrOx film annealed at 200 °C [24]. The inset in Figure 2 depicts the MIM (Mo/HfZrOx/Mo) structure.

Figure 3a–c presents the surface morphology of HfZrO-200 °C, HfZrO-250 °C, and HfZrO-300 °C films. The root-mean-square roughness (R_RMS_) values for the HfZrO_x_ films annealed at 200 °C, 250 °C, and 300 °C are determined to be 0.57 nm, 0.36 nm, and 0.29 nm, respectively, with a scanning area of 2 μm × 2 μm. With increasing annealing temperature, the surface roughness decreases, resulting in a smoother surface morphology [20]. The improved smoothness of the HfZrO_x_ film surface enhances the interface quality with the channel layer. Consequently, the favorable interface between the gate insulator and the channel layer contributes to the enhanced electrical properties of the LaZnO TFT [2,8,12,20,24,25,26,29,30].

The O1s XPS spectra were analyzed to investigate the chemical composition of the HfZrO films (Figure 4a–c). The O 1s peak was deconvoluted into three sub-peaks centered around ~529.5 eV (metal oxide, M-O), ~530.5 eV (oxygen vacancy, Vo), and ~532 eV (hydroxyl groups, -OH). The percentages of M-O-M, Vo, and -OH in the HfZrOx films annealed at different temperatures (200 °C, 250 °C, and 300 °C) were determined. The M-O-M percentages were found to be 64.75%, 75.39%, and 77.65%, while the Vo ratios were 24.75%, 17.28%, and 15.98%, and the -OH ratios were 10.50%, 7.33%, and 6.37%, respectively [2,15]. The higher M-O-M content indicates a reduction in oxygen-related defects (Vo + -OH), resulting in fewer defects at the interface between the gate insulator and channel layers. This contributes to improved device performance and interface quality [5,10,14,15,27,31,32,33]. 

The electrical properties of LaZnO TFTs with HfZrO (HfZrO-200 °C, HfZrO-250 °C, and HfZrO-300 °C) gate insulators were studied by measuring the I-V (transfer) curves of TFTs. Figure 5a–c depicts the transfer curves of LaZnO TFTs with HfZrO-200 °C, HfZrO-250 °C, and HfZrO-300 °C, illustrating their hysteresis characteristics, with corresponding electrical properties, such as μ_FE_ of (8.49, 19.06, and 22.28) cm^2^V^−1^s^−1^, V_TH_ of (2.60, 1.98, and 1.87) V, V_H_ of (0.03, 0, and 0) V, and SS of (295, 256, and 231) mV/dec, respectively. The hysteresis curves for the LaZnO TFTs with HfZrO-200 °C, HfZrO-250 °C, and HfZrO-300 °C show the negligible V_H_, indicating a favorable gate insulator (GI)/channel interface. The reduced number of traps at the GI/channel interface contributes to higher I_ON_ and lower SS, as shown in Figure 5a–c. Achieving a higher M-O-M ratio and minimizing oxygen-related defects leads to smoother charge transport and reduced charge trapping, resulting in enhanced carrier mobility and stable device operation. With an increase in HfZrO_x_ annealing temperature, the V_TH_ shifts to a positive V_GS_ due to a decrease in trap states at the interface [2,10,24,33]. The output curves of LaZnO TFTs with HfZrO-200 °C, HfZrO-250 °C, and HfZrO-300 °C are presented in Figure 6a–c, exhibiting clear pinch-off and saturation behavior. The TFTs were tested with V_DS_ sweeping from 0 to +5 V and V_GS_ sweeping from 0 to +5 V (step = 1 V). The absence of current crowding in the low V_DS_ region confirms excellent ohmic contact between the source/drain electrodes and channel layers [5,10,16,29].

Figure 7a presents an optical photograph of the measurement setup for the flexible LaZnO TFT fabricated on a PI substrate with a HfZrO-250 °C gate insulator (GI). Figure 7b displays the transfer curve of the flexible LaZnO TFT, measured at V_DS_ of 0.1 V by sweeping V_GS_ from −5 to +5 V. The I_DS_^1/2^ vs. V_GS_ plot with a linear extrapolation line is shown on the right side of the *y*-axis. The flexible LaZnO TFT with HfZrO-250 °C GI exhibits a μ_FE_ of 20.55 cm^2^V^−1^s^−1^, V_TH_ of 1.12 V, and SS of 264 mV/dec. Compared with TFTs on glass substrates, the electrical properties of the flexible LaZnO TFT show almost negligible changes. To assess the bias stability of the LaZnO TFT with HfZrO-250 °C GI, PBTS was performed on the TFT (at V_GS_ = 5 V for 1 h), as shown in Figure 7c. The evaluation of the transfer curve under PBTS reveals a threshold voltage shift (ΔV_TH_) of 0.25 V. The positive shift in V_TH_ is attributed to electron trapping at the interface between the HfZrO and LaZnO layers. However, the SS of the LaZnO TFT undergoes a negligible change after 1 h of bias stress, indicating fewer interfacial traps. A higher M-O-M ratio and reduced oxygen-related defects significantly enhance mobility and excellent bias stability of the metal oxide TFT. Thus, the interface quality between HfZrOx and LaZnO plays a crucial role in improving the electrical properties of the TFT.

To validate the accuracy of the performances of HfZrO-250 °C/LaZnO TFTs, we fabricated the TFTs in four different runs to ensure precise and reliable data collection. The m_FE_, V_ON_, and SS values of the HfZrO-250 °C/LaZnO TFT for Run-I, Run-II, Run-III, and Run-IV are (17.50 ± 2.20, 16.95 ± 2.85, 18.99 ± 2.15, and 18.50 ± 1.98) cm^2^V^−1^s^−1^, (−1.62 ± 0.18, −1.56 ± 0.14, −1.72 ± 0.21, and −1.69 ± 0.23) V, (253 ± 13.22, 264 ± 11.94, 249 ± 10.89, and 261 ± 8.58) mV/dec, respectively, as shown in Figure 8a–c. The small deviation in these values indicates the high reliability and consistency of the fabrication process. The narrow range of fluctuations highlights the robustness of the HfZrO-250 °C/LaZnO TFT performance across multiple runs, reaffirming its potential for practical applications in electronic devices. The HZO film deposited at 350 °C is prone to crystallization and can be a ferroelectric material [33]. As a result, we used spin-coating for the HZO film at 250 °C, which provides a more favorable gate insulator for the LaZnO TFT. The electrical properties of the solution-processed ZnO-based TFTs with various solution-processed gate dielectrics are reported in the literature, as shown in Table 1 [34,35,36,37,38,39]. H. Liu et al., demonstrate hysteresis-free Indium gallium zinc oxide (IGZO) TFT with HfOx gate insulator (GI), which exhibits excellent performance characteristics, including a high mobility of 30 cm^2^V^−1^s^−1^, an SS of 68 mV/dec, and a high I_ON_/I_OFF_ of 10^6^ [40]. In the present work, our HfZrO/LaZnO TFTs display superior performance with high mobility, low subthreshold swing, and a high I_ON_/I_OFF_ ratio. Furthermore, we successfully fabricated flexible LaZnO TFTs with HfZrO-250 °C GI, which exhibited hysteresis-free behavior, leading to improved electronic device performance and enhanced stability for a wide range of applications [34,35,36,41].

## 4. Conclusions

In summary, this study investigates the use of a low-temperature solution-processed HfZrOx gate insulator (GI) to improve the performance of LaZnO thin-film transistors (TFTs) for potential applications in next-generation flexible displays. To optimize the HfZrO GI, annealing was performed at 200, 250, or 300 °C. The LaZnO TFT with HfZrO-250 °C GI exhibits a μ_FE_ of 19.06 cm^2^V^−1^s^−1^, V_TH_ of 1.98 V, hysteresis voltage (V_H_) of 0 V, SS of 256 mV/dec, and I_ON_/I_OFF_ ratio of ~10^8^. The decrease in hysteresis voltage is critical as it helps to ensure the stability and reliability of the TFT operation. Additionally, the steep subthreshold slope (SS) of 256 mV/dec indicates a sharp turn-on behavior, which is essential for achieving efficient switching. Moreover, the on/off current ratio (I_ON_/I_OFF_) of ~10^8^ highlights the ability of the transistor to efficiently control the flow of currents. Under PBTS, the HfZrO-250 °C/LaZnO TFT experiences a threshold voltage shift (ΔV_TH_) of 0.23 V. These enhancements can be attributed to the smooth surface morphology and reduced defects in the HfZrO gate insulator. Therefore, these findings significantly advance our understanding of the underlying phenomena and pave the way for the development of flexible metal-oxide-semiconductor TFTs for future-generation flexible displays.

## Figures and Tables

**Figure 1 nanomaterials-13-02410-f001:**
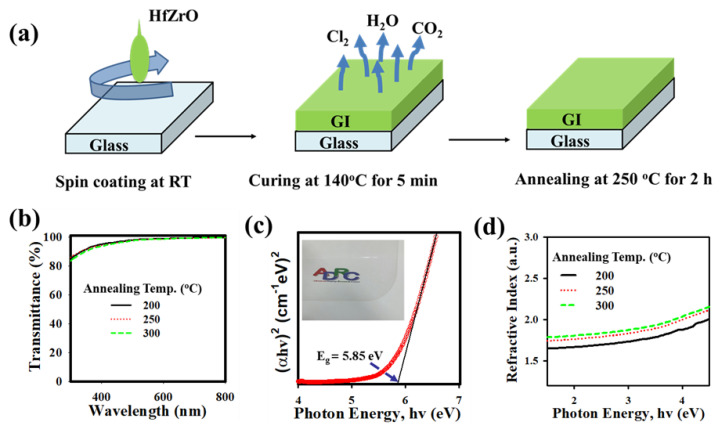
(**a**) Schematic representation of HfZrO_x_ thin film deposited by spin coating. (**b**) The transmittances of HfZrO-200 °C, HfZrO-250 °C, and HfZrO-300 °C thin films are higher than 90% in the visible region. (**c**) The bandgap of HfZrO-250 °C thin film obtained from the Tauc plot. The inset of (**c**) shows a photograph of HfZrO-250 °C thin film placed on the ADRC logo. (**d**) The refractive index as a function of the photon energy of HfZrO-200 °C, HfZrO-250 °C, and HfZrO-300 °C thin films.

**Figure 2 nanomaterials-13-02410-f002:**
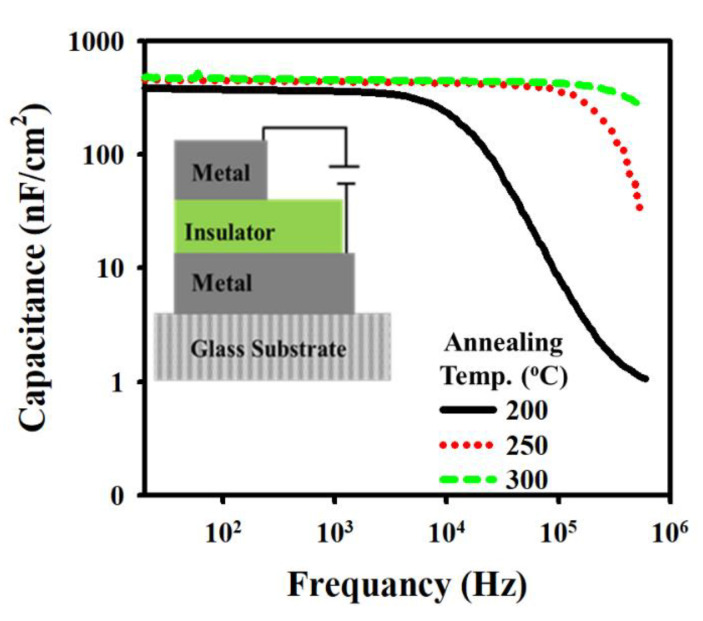
Capacitance vs. frequency of HfZrO-200 °C, HfZrO-250 °C, and HfZrO-300 °C gate insulators (GIs) annealed at different temperatures.

**Figure 3 nanomaterials-13-02410-f003:**
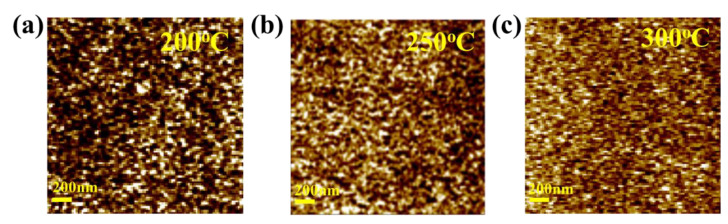
AFM images (Scan size 2 μm × 2 μm) of (**a**) HfZrO-200 °C, (**b**) HfZrO-250 °C, and (**c**) HfZrO-300 °C thin films. The R_RMS_ values of 0.57, 0.36, and 0.29 nm indicate smooth surface morphology.

**Figure 4 nanomaterials-13-02410-f004:**
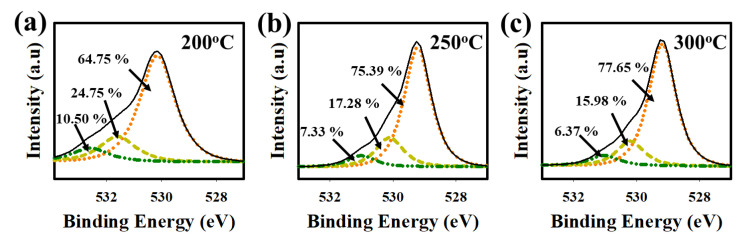
Deconvoluted O1s spectra of (**a**) HfZrO-200 °C, (**b**) HfZrO-250 °C, and (**c**) HfZrO-300 °C thin films. O1s spectra are deconvoluted into three sub-peaks (M-O, Vo, and -OH), where M = metal, Vo = oxygen vacancy, -OH = hydroxyl group.

**Figure 5 nanomaterials-13-02410-f005:**
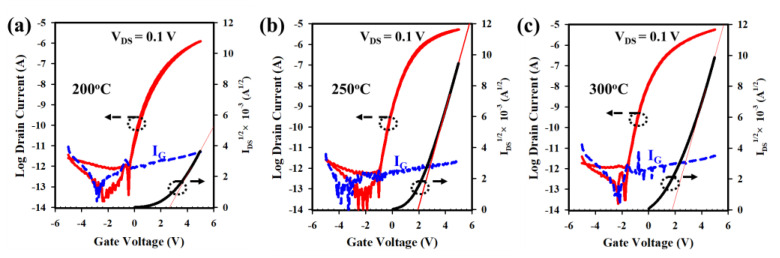
(**a**–**c**) The transfer characteristics with hysteresis curves for the LaZnO TFTs with HfZrO-200 °C, HfZrO-250 °C, and HfZrO-300 °C GIs. Hysteresis voltage was obtained at I_DS_ = 10^−10^ A. I_G_ represents the gate leakage current.

**Figure 6 nanomaterials-13-02410-f006:**
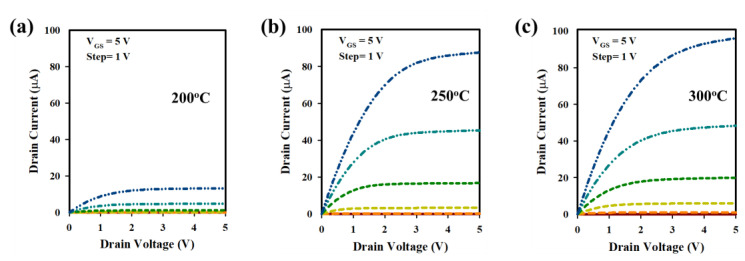
(**a**–**c**) The output curves of LaZnO TFTs with HfZrO-200 °C, HfZrO-250 °C, and HfZrO-300 °C GIs. The channel width and length of the LaZnO TFT used for the measurement are 20 and 10 μm, respectively.

**Figure 7 nanomaterials-13-02410-f007:**
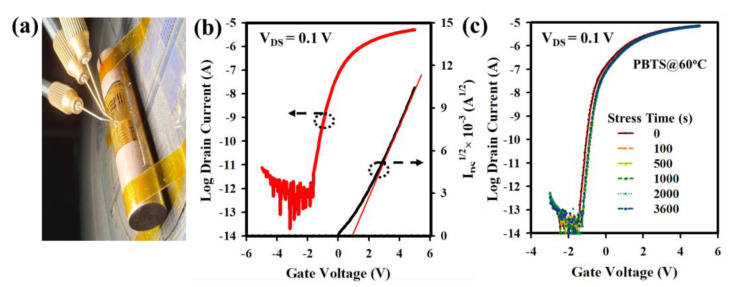
(**a**) Photograph of the measurement setup of LaZnO TFT with HfZrO-250 °C gate insulator fabricated on a PI substrate. (**b**) The transfer curve of the LaZnO TFT measured at the drain voltage V_DS_ = 0.1 V by sweeping V_GS_ from −5 to +5 V. (**c**) Transfer curve of the LaZnO TFT under PBTS for 1 h at 60 °C.

**Figure 8 nanomaterials-13-02410-f008:**
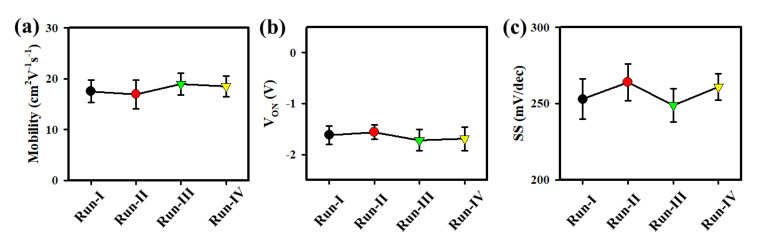
(**a**) Mobility (**b**) V_ON_, and (**c**) SS of HfZrO-250 °C/LaZnO TFT as a function of run (Run-I, Run-II, Run-III, and Run-IV). All the TFTs were measured at room temperature in the dark.

**Table 1 nanomaterials-13-02410-t001:** Comparison of the electrical properties of the solution-processed ZnO-based TFTs with various solution-processed gate dielectrics reported in the literature.

Active/GI	TFT W/L [μm/μm]	Mobility [cm^2^V^−1^s^−1^]	SS [mV/dec]	Stability (ΔV_TH_, [V])	Ref.
^(a)^ LaZnO/^(b)^ ZrO_x_	50/10	8.31	218	PBTS (0.10)	[2]
^(b)^ ZnO/^(b)^ AlO_x_	60/10	6.05	550	-	[34]
^(a)^ ZnO/^(a)^ AlTiO_x_	2000/20	10.00	550	-	[35]
^(a)^ ZnO/^(a)^ HfO_x_	2000/20	42.00	-	-	[36]
^(b)^ ZnO/^(b)^ LaZrO_x_	50/10	11.58	249	PBS (0.20)	[37]
^(b)^ ZnO/^(b)^ SiO_2_	1000/50	3.20	600	-	[38]
^(a)^ ZnO/^(b)^ ZrO_x_	50/10	12.76	260	PBS (0.01)	[39]
^(a)^ LaZnO/^(b)^ HfZrO	50/10	19.06	256	PBTS (0.23)	[This work]

^(a)^ Spray Pyrolysis; ^(b)^ Spin Coating.

## Data Availability

We can provide the data if asked.

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
