# Peer review of "Low-Temperature Solution-Processed HfZrO Gate Insulator for High-Performance of Flexible LaZnO Thin-Film Transistor"

_nanomaterials, 2023, doi:10.3390/nano13172410_

Round 1
Reviewer 1 Report
The authors have continuously demonstrated high quality work in the field of complex oxides TFTs. This work on low-temperature solution processed HfZrO gate insulator for LaZnO TFT is yet another good work in that direction. The authors report that the smooth surface morphology and the lower defect density resulting from a low temperature fabrication process is responsible for negligible change in VTh under PBTS.
The article is very interesting and should be published.
However, the article does not mention anything about the stoichiometry of any of the complex oxides. Would the authors be able to give any information on the stoichiometry of the compounds ?
Reviewer 2 Report
In this work, a low-temperature solution-processed HfZrOx gate insulator (GI) is presented, for improving device performance of the LaZnO TFTs. The device performance is optimized in the HfZrO-250 C GI-based LZO TFT, with a μFE of 19.06 cm2V-1s-1, threshold voltage (VTH) of 1.98 V, hysteresis voltage (VH) of 0 V, subthreshold swing (SS) of 256 mV/dec., and ION/IOFF of ~108.
However, the authors should be very cautious to use ‘VH = 0V’, in the manuscript.
Besides, other errors or comments are as following:
1 inappropriate literature citation: As 32 articles listed in the References, an inappropriate citation was done in the 1st and 2nd paragraphs of Introduction, escaping from 24 to 30. Etc..
2 inaccurate physical definition: as illustrated in equation (2), SS= d(VGS)/d(log log IDS) .
3. While investigating the stability of the LaZnO TFT with HfZrO-250 C GI, a small threshold voltage shift (ΔVTH) of 0.25 V was achieved in Fig. 7c, for the PBTS measurements performed at VGS = 5 V for 1 h and with temperature of 60 C. However, while comparing the transfer curves of device worked at room temperature and 60C, there is an obvious shift from Fig.7b to 7C, there is not any explanation.
4. In table I, for the figure-of-merit, it is not appropriate to compare the device performances with various fabrication technologies, among the sputtering and spin coating, and so on.
5. there is an obvious mistake of marking, in Fig.3C, with 300C.
6. What do the blue curves in Fig. 5 represent? the gate leakage current?
7. The authors concluded that, the enhancements in the LaZnO TFT with HfZrO-250 C GI were attributed to the smooth surface morphology and reduced defect density achieved with the HfZrO gate insulator. Other characterizations, like low-frequency noise spectra, are suggested to be performed.
the English writing is good, in this manuscript.
Reviewer 3 Report
1. The abstract is simply written. Therefore, Important finding should be mentioned in the abstract.
2. Complete the abstract with possible applications.
3. You highlight novelty aspects of literature
4. The authors should formulate very briefly in the introduction the novelty of the work with respect to the other works in the field.
5. How the present author's results differ from the earlier reported ones? Give a comparison of the results and discussion part.
6. The conclusion is just a list of observations like for a technical report. Please add the science including for cause of the effects in order to advance the understanding of the studied phenomena compared to what is already known or could be expected from literature.
7. The accuracy of the measurements of the technology should be presented.
8. In summary, kindly include the advantage of the present study over other studies.
Reviewer 4 Report
The authors demonstrate low temperature processing of LaZnO/HfZrOx solution processed transistors for their applications in flexible thin-film transistors. However, there are several issues that are not addressed and cannot be considered for publication in Nanomaterials in its present form. Below are the comments towards the manuscript:
1. The authors should clearly mention the novelty of their work, as the group has previously demonstrated optimized results of the individual LaZnO semiconductor and HfZrO dielectric as per reference 2 and 24 in the manuscript. Also, recent work on spin on HfOx spin coated dielectrics with optimized precursor formulation have demonstrated impressive results at a processing temperature of 200C (DOI: 10.1109/LED.2023.3295594)
2. Why have the authors chosen the 250C sample for Flexible substrates, rather than 300C or higher samples, since PI can handle fabrication temperatures of 300C? Especially, since their previous work on the same HZO dielectric, (Ref 24) was optimized for best performance at 300C and the LaZnO precursor did not decompose sufficient well at 250C and needed 350C for complete decomposition, as indicated by the group’s previous work (Ref 2).
3. Can the authors comment on why the HZO was spin coated and not spray coated like the LaZnO, since spray coating enables a higher film density?
4. In figure Table1, authors claim that their LaZnO is fabricated via sputtering, but is prepared by solution processing as described in the experimental section?
5. The labelling of the Lanthanum doped zinc oxide is inconsistent throughout the manuscript, (LZO and LaZnO). Please resort to a single abbreviation to avoid confusion. Also, Figure 3c is incorrectly labelled as 250C
Unfortunately, due to the incremental nature of the work and lack of novelty in the manuscript, I cannot recommend the manuscript for publication in Nanomaterials.
Round 2
Reviewer 4 Report
I am satisfied with the detailed responses provided by the authors in the revised manuscript and clearly highlighting the novelty of their work. I can now recommend the manuscript for publication in Nanomaterials.